# Redefined ion association constants have consequences for calcium phosphate nucleation and biomineralization

David P. McDonogh [1], Julian D. Gale [2], Paolo Raiteri [2] ✉ & Denis Gebauer [1] ✉

Calcium orthophosphates (CaPs), as hydroxyapatite (HAP) in bones and teeth are the most important biomineral for humankind. While clusters in CaP nucleation have long been known, their speciation and mechanistic pathways to HAP remain debated. Evidently, mineral nucleation begins with two ions interacting in solution, fundamentally underlying solute clustering. Here, we explore CaP ion association using potentiometric methods and computer simulations. Our results agree with literature association constants for $Ca^{2+}$ and $H_2PO_4^-$, and $Ca^{2+}$ and $HPO_4^{2-}$, but not for $Ca^{2+}$ and $PO_4^{3-}$ ions, which previously has been strongly overestimated by two orders of magnitude. Our data suggests that the discrepancy is due to a subtle, premature phase separation that can occur at low ion activity products, especially at higher pH. We provide an important revision of long used literature constants, where association of $Ca^{2+}$ and $PO_4^{3-}$ actually becomes negligible below pH 9.0, in contrast to previous values. Instead, $[CaHPO_4]^0$ dominates the aqueous CaP speciation between pH ~6–10. Consequently, calcium hydrogen phosphate association is critical in cluster-based precipitation in the near-neutral pH regime, e.g., in biomineralization. The revised thermodynamics reveal significant and thus far unexplored multi-anion association in computer simulations, constituting a kinetic trap that further complicates aqueous calcium phosphate speciation.

Calcium orthophosphate (CaP) phases are arguably the most important mineral group for mankind, making up bones and teeth, coatings on medical implants, and ingredients in foods[1–4]. Technical applications of CaPs have also long been studied resulting in various uses in protein chromatography, waste water treatment, fertilisers and catalysis[5–9]. There are more than 10 known pure solid CaPs, and additional phases are still being actively discovered[10,11]. Facile substitution of foreign ions, as in fluorapatite, for example, further increases the variety in this system. Despite such widespread relevance of these minerals, the formation mechanisms, structure and even compositions of the precipitated solids are still debated in the literature, both synthetic and biological[12–16]. The role of amorphous CaP (ACP), as a precursor to crystalline phases, especially has been an active area of research[17–22].

Association between two ions is fundamentally the first step toward nucleation, regardless of whether a classical or non-classical view is taken. In the classical model, single ions are added in a stepwise manner, at first forming unstable pre-critical nuclei, growing into a critically-sized metastable nucleus as a transition state that can proceed to form a crystal[23]. In the non-classical pre-nucleation cluster (PNC) pathway, associated ions form stable, solute clusters[24]. Phase separation subsequently occurs via a reduction in dynamics of the clusters, corresponding to nanoscopic liquid-liquid de-mixing, successive aggregation and de-solvation/dehydration toward the

[1]Institute of Inorganic Chemistry, Leibniz University Hannover, Callinstr. 9, 30167 Hannover, Germany. [2]Curtin Institute for Computation and School of Molecular and Life Sciences, Curtin University, P.O. Box U1987, Perth, WA 6845, Australia. ✉e-mail: p.raiteri@curtin.edu.au; gebauer@acc.uni-hannover.de

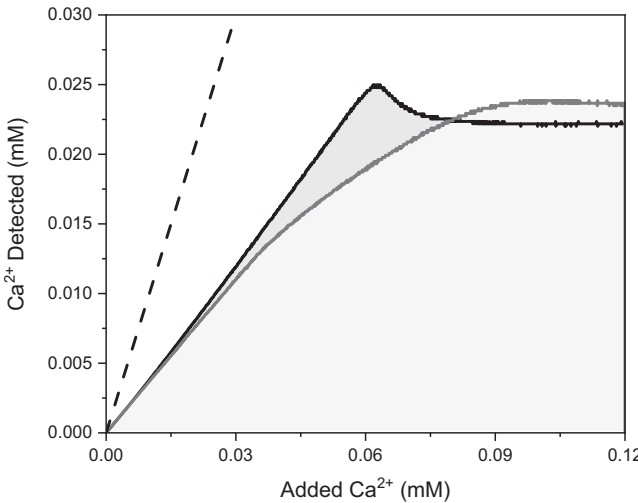

**Fig. 1 | Two titrations of Ca²⁺ ions into phosphate solution at pH 9.8 with strikingly different profiles.** The black curve uses the same experimental conditions as the dark grey curve, except that the calcium solution contains 0.05 M HCl (full data Supplementary Fig. 1). The dashed line represents the concentration of calcium expected if no binding occurs; both titration curves lie below this due to the ion association between Ca²⁺ and phosphate ions. The acidified titration (black) shows a higher amount of free calcium ions in the solution and remains linear for a greater amount of added calcium ions (grey and light grey shading highlight the relevant areas under the curves). Additionally, the acidified experiment shows the expected nucleation profile, increasing up to a peak before dropping to a plateau determined by the solubility of the precipitated phase, i.e., with supersaturation. The non-acidified experiment (dark grey) bends early in the titration, and takes a non-linear path directly to the plateau, i.e., without apparent supersaturation. This is caused by a premature phase separation at the dosing tip due to a local over-concentration of calcium (Supplementary Section 1, Supplementary Fig. 2). Sequestration of calcium ions in a separated phase leads to perceived higher ion association.

formation of solid amorphous intermediates[25]. Although the introduction of the PNC model for calcium carbonate has ignited controversy, multiple experimental methods support the existence of these clusters[26,27]. Larger poly-anion/cation species have also been discussed for iron (oxy)(hydr)oxides among others[28]. A Ca₉(PO₄)₆ cluster has indeed long been established in the CaP system, proposed by and named after Posner in the 1970s[29,30]. However, the apparent agreement in size between Posner's formula and entities detected in solutions may also be consistent with alternate cluster compositions and structures as well[31]. The structure, composition and speciation of pre-nucleation species and their significance in nucleation are still under investigation, particularly their role in biological systems[32–35].

With the renewed interest in the early stages of precipitation mechanisms, understanding the ion interactions before macroscopic solids are formed has become especially important from a non-classical point of view[36–38]. The binding between different ions in solution, however, has been studied since the early days in chemistry, since these equilibria determine solution composition and characteristics, as evident from the extensive compilations of ion association constants published in the last century[39,40]. Taking a broad view, the most fundamental parameter of aqueous chemistry, pH, is also essentially a description of ion binding. For the CaPs, the different ion associates have been shown to be vital in understanding the solubility of solid phases[41,42]. Current research investigates the role of these associated species in nucleation mechanisms.

Recent developments in potentiometric and cryo-TEM methods have enabled investigations into nucleation; however, simulations have remained the only tool capable of investigating the atomistic details of the initial stages of phase separation. Combining these

methods, Habraken et al. have proposed pre-nucleation [Ca(HPO₄)₃]⁴⁻ complexes that aggregate to form polymeric strands early in the nucleation mechanism[31]. According to their model, nucleation would proceed via the uptake of additional calcium ions forming species with a base formula of [Ca₂(HPO₄)₃]²⁻. These results, among others, however, were challenged by Garcia et al. who predicted that the ion pair and smaller aggregation units would be more stable in solution, casting doubt on the existence and relevance of the proposed [Ca(HPO₄)₃]⁴⁻ pre-nucleation complex[43]. The theoretical model used, specifically a force field description of the energy and forces, was developed by the same group and extensively validated against experimental and ab initio results[44]. The ion association constants used to benchmark the predictions of this model can be traced to Chughtai et al. in 1968 (see Supplementary Table 1 for a full list of literature values)[45]. This set of ion pairing constants has become the standard for any mathematical description of the CaP system, as this is nearly the only complete set available in the literature. Obviously, it is vital that the experimental values for association constants used as either input to chemical speciation calculations, or to validate simulation models, are sound; otherwise, the calculated speciation will be erroneous or the need for more accurate simulation models may go undetected.

According to the commonly used ion association constants, the $PO_4^{3-}$ anion binds a Ca²⁺ cation four orders of magnitude more strongly than the $HPO_4^{2-}$ anion[45]. This is quite an increase considering that $HPO_4^{2-}$ and $CO_3^{2-}$ ions bind a calcium ion around two orders of magnitude more strongly than $H_2PO_4^{-}$ and $HCO_3^{-}$ ions, respectively[37,40,45]. In the $PO_4^{3-}$ ion, the charge is delocalised across all four oxygen atoms – the Ca²⁺ ion thus does not see a triply negative point charge, while in solution the strength of the Ca-PO₄ interaction will be diminished by the screening effect of the solvent, as well as the competition with the substantially more exergonic hydration free energy of the fully deprotonated phosphate anion. Considering this, the third ion association constant reported in the literature seems disproportionally large.

Here, we provide experimental evidence for a subtle phase separation event occurring at very low ion activity products, most probably having biased previous ion association constant determinations, especially for Ca²⁺ and $PO_4^{3-}$ ions (see Fig. 1 and relevant discussion in Results). Below, we re-evaluate the pre-nucleation ion association for the CaPs via advanced titration methods, thereby circumventing the above-mentioned issue of a convoluted measurement of ion association and phase separation and assessing pure ion association. Based on this, we propose revised values for the binding constants that indicate a substantial reduction in the binding between Ca²⁺ and $PO_4^{3-}$ ions. This finding of a lower increase in the ion pairing when going from $HPO_4^{2-}$ to $PO_4^{3-}$ is supported by both quantum mechanical calculations and simulations using an improved model that includes the important effects of polarisation.

## Results

To study the ion association in the calcium-phosphate system we adapted a titration method developed for CaCO₃ nucleation experiments[37]. As demonstrated in Fig. 1, when adding calcium chloride solution to a phosphate buffer the measured concentration of free calcium ions is less than the added amount. Ion association between the calcium and various phosphate species ($H_2PO_4^{-}$, $HPO_4^{2-}$ and $PO_4^{3-}$) determines the difference between the two values.

As demonstrated in Fig. 1, simply adding a Ca²⁺ containing solution to phosphate ions leads to a non-linear curve, progressing directly to a plateau, without the characteristic linear pre-nucleation regime and peak at nucleation[13,37]. A local overconcentration of calcium at the dosing tip causes a CaP phase to form (Supplementary Fig. 2); this precipitated phase sequesters ions which otherwise would be in solution, increasing the amount of calcium bound, especially as pH increases. Any calculations of the ion association using such data then

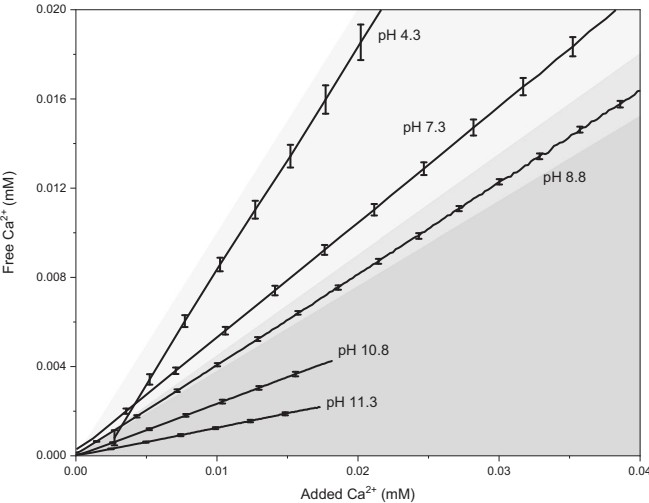
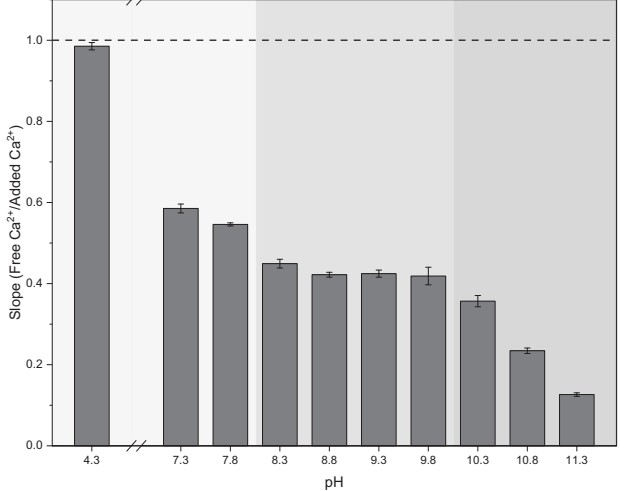

**Fig. 2 | Left: Development of free calcium in a phosphate buffer with the slow addition of calcium chloride stock solution across a range of pH values.** Curves at selected pH values are shown as averages of three identical measurements; error bars are ± the standard deviation ($N = 3$), only shown for a few data points for clarity. The transition from white to grey shows the theoretical maximum calcium in solution, i.e., the calcium ion concentration for zero ion association. All of the measured curves lie below this line, with the deviation increasing as the pH is raised.

Three pH regimes are shown: light grey, low pH ($\approx 4–8$); mid-range ($\approx 8–10$); and high pH (above 10). See Supplementary Figs. 3, 4 for the full data. Right: Slope (i.e., ratio of free $Ca^{2+}$ to added $Ca^{2+}$) of the pre-nucleation titration curve at measured pH values; values are averages of three experiments and error bars ± the standard deviation ($N = 3$). The dashed line represents the maximum amount of free $Ca^{2+}$. The grey shading indicates the pH ranges from the left figure.

overestimate the homogeneous binding constants. This is likely the case for the $Ca^{2+}$ and $PO_4^{3-}$ interaction in the literature. Simply acidifying the added $Ca^{2+}$ solution solves this problem; the lower local pH increases the solubility of solid phases – the ions are directly distributed into the solution and the ion association at the desired pH can be observed. The black curve in Fig. 1 develops as expected for such a titration, in that there is a linear pre-nucleation regime and the grey shading highlights the amount of $Ca^{2+}$ missing from the solution. Using this modified titration method, the pre-nucleation regime can be investigated over a wide pH range, as summarised in Fig. 2 (for details, see Supplementary Figs. 3 and 4).

There is a noticeable pH dependence in the curves displayed in Fig. 2. As pH increases, the linear binding regime becomes shorter as precipitation occurs at lower amounts of added calcium. At this point, solubility equilibria begin to dominate, and ion association can no-longer be evaluated. The curves have been truncated just before nucleation of solids occurs, as the post-nucleation regime is highly complex and beyond the scope of this work. Additionally, the slopes of the curves decrease with increasing pH as more calcium ions are bound as the phosphate ions become progressively less protonated.

Using a similar experimental setup, in combination with analytical ultracentrifugation, it has been shown that the formation of pre-nucleation clusters in the calcium carbonate system cannot be distinguished from ion pair formation – more precisely, the mathematical description using the law of mass action yields linear binding profiles in excess carbonate buffer in both cases[37,46]. More concisely, linear pre-nucleation regimes similar to those in Fig. 2 do not exclude multiple binding[47]. The same can be assumed for the CaPs and finds precedent in the literature. While various pre-nucleation species have been proposed for this system since the 1970s, all treatments of ion association have been based on law of mass action calculations for the formation of ion pairs from individual ions[31]. Using the same basis ensures that the analysis here is comparable to earlier works, while the values obtained likely also characterize the formation of larger pre-nucleation entities[37,46]. Indeed, our computational studies indicate that this may also include multi-anion ($HPO_4^{2-}/HPO_4^{2-}$) association, possibly constituting a so far largely unexplored additional contribution to cluster formation in the CaP system (see Discussion).

Assuming there is only 1:1 ion association, the observed calcium ion binding can be explained based on the following equilibria:

$$Ca^{2+} + H_2PO_4^- \rightleftharpoons \left[CaH_2PO_4\right]^+ \tag{1}$$

$$Ca^{2+} + HPO_4^{2-} \rightleftharpoons \left[CaHPO_4\right]^0 \tag{2}$$

$$Ca^{2+} + PO_4^{3-} \rightleftharpoons \left[CaPO_4\right]^- \tag{3}$$

The constants, $K_i$, for the interaction between calcium and the various phosphate ions are then defined in terms of ion activities ($a_i$) using the law of mass action:

$$K_1 = \frac{a_{\left[CaH_2PO_4\right]^+}}{a_{Ca^{2+}_{free}} \, a_{H_2PO_4^-}} \tag{4}$$

$$K_2 = \frac{a_{\left[CaHPO_4\right]^0}}{a_{Ca^{2+}_{free}} \, a_{HPO_4^{2-}}} \tag{5}$$

$$K_3 = \frac{a_{\left[CaPO_4\right]^-}}{a_{Ca^{2+}_{free}} \, a_{PO_4^{3-}}} \tag{6}$$

In the experiments above, the calcium concentration was measured directly, allowing the concentration of other relevant species to be calculated[48]. These values are converted to activities for the ion association calculation. This is done by determining the ionic strength of the solution and then applying the Davies equation for the individual species (see Methods and Supplementary Figs. 5, 6)[49].

Analysis of the pre-nucleation ion association is used to determine the $K_i$ values. It must be noted that the ion association studied here is a purely thermodynamic process and kinetic factors such as addition rates do not influence the binding data used to evaluate the equilibria behind ion pair formation (Supplementary Figs. 7 and 8). Although similar titration experiments have been used to investigate nucleation kinetics and to assess the effects of addition rates on post-nucleation

**Table 1 | Overview of the activity-based ion association constants**

|  | $K_1$ | $K_2$ | $K_3$ |
|---|---|---|---|
| Ion Pairing Constant | 4.5 | 470 | 50,000 |
| Error | ± 2.4 | ± 50 | ± 8000 |

The values for each of the three possible ion associates between calcium and phosphate ions as calculated from the titration data. The reported values show where agreement between calculation and experiment is optimal, while the errors provide the bounds to the reasonable range where the ion pairing constants lie. The error is largely due to the global fitting of all pH values; the deviation between curves within one set of measurements is generally less than 5%, and therefore relatively insignificant for the evaluation of ion association shown here. The limits where the calculated curves begin to deviate from the experimental data are explored in the Supplementary Information (Supplementary Figs. 9–11).

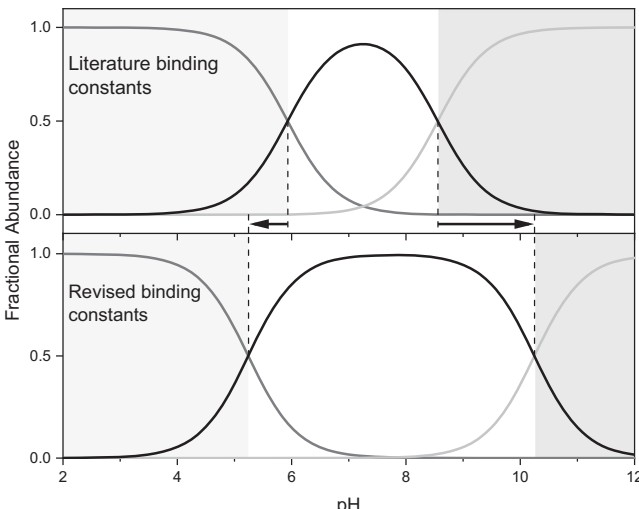

**Fig. 3 | Speciation of CaP ion pairs from pH 2 to 12 calculated using PHREEQC.** The curve for $[CaHPO_4]^0$ is black in the middle, while $[CaH_2PO_4]^+$ and $[CaPO_4]^-$ are grey on the left and right, respectively[96,97]. The top figure uses the ion association constants from Chughtai et al. while the bottom figure uses the constants from this work[45]. The effect of the large literature third ion association constant is immediately obvious; the $[CaPO_4]^-$ ion pair is dominant over a much larger pH range. The crossover to other calcium-phosphate ion pairs moves down about one pH unit and up by about two units at low and high pH values, respectively. Note that the mole fraction of each species is based on the total amount of ion pairs in solution; especially at low pH a large mole fraction does not imply a high concentration, only that a large fraction of the total ion pairs exist as a given species. See Supplementary Figs. 13, 14 for the percentage of bound calcium across the same pH range, and how this correlates with the speciation of phosphate ions.

behaviours, the present study is strictly within the pre-nucleation regime, which is governed by thermodynamics[46,50]. Here, two models are developed; one incorporating experimentally measured calcium concentrations (Direct Calculation), the other predicting solution composition purely from pH, activity coefficients and the amount of added calcium (Predictive Calculation). The value for $K_1$ can be directly calculated using the law of mass action at pH 4.3 as $H_2PO_4^-$ dominates at this pH. The $K_i$ values were found to be 4.5, 470 and 50,000; the ion association constants from experiment and titration are compared in Table 1 and in Supplementary Figs. 9–11 (further discussion in Supplementary Section 2).

As seen in Supplementary Fig. 9, the binding is slightly underestimated at the highest pH values and it is overestimated at lower pH. In-between these extremes the binding is overestimated (pH 10.3 and 9.8), underestimated (pH 9.3 and 8.8), and nearly spot on (pH 8.3). In all cases the Predictive Calculation lies closer to the experimental data, and the curves are straight lines. Meanwhile the Direct Calculation model mirrors curvature in the data at pH 7.3 and 7.8, while also deviating from the measured curves at a pH of 11.3.

The value for $K_1$ measured in this work accords well with the more recent literature values from either Gregory et al. or Zhang et al.; some authors, including Chughtai et al. report higher values, however, the majority are near our value (see Supplementary Table 1)[45,51,52]. $K_2$ falls in the middle of the spread from ca. 200 to 600[45,51,52]. The third association constant, $K_3$, is two orders of magnitude lower than the commonly used value. The deviation in the amount of bound $Ca^{2+}$ in the titration between experimentally measured and modelled values when using the literature values is shown in Supplementary Fig. 12.

While $K_1$ and $K_2$ can be measured at pH values where the respective phosphate species are dominant in solution, the large third dissociation constant of phosphoric acid causes $PO_4^{3-}$ only to be present in significant amounts at high pH values, making $K_3$ difficult to determine, due to the properties of the electrochemical sensors. Literature values for $K_3$ are scarce, likely due to difficulties in achieving accurate measurements at the required high pH values and low concentrations, making a rigorous comparison difficult. The $PO_4^{3-}$ ion accounts for roughly 0.95, 2.9 and 8.7% of the total phosphate in solution at pH 10.3, 10.8 and 11.3, respectively (based on $pK_a$ values measured in the titration setup). While the fractional abundance does not exceed 10% at the experimental conditions, the large association constant means that the influence of this species can be measured, similar to the case of calcium carbonate versus bicarbonate[37]. A pH level above 12.5 would be required for $PO_4^{3-}$ to dominate the speciation, at which the electrodes are unreliable. Considering the errors generally associated with this constant (Supplementary Table 1), the $K_3$ value is comparable to that of Atlas et al., however, very high ionic strengths were used in their work[53].

The $K_3$ values reported here, and by both Chughtai et al. and Zhang et al. were determined using titration based methods[45,52]. As discussed more thoroughly above and in the Supplementary

Information (Supplementary Section 1), acidification of the added calcium solution is necessary to avoid premature phase separation due to increased local supersaturation when calcium is added to dilute phosphate buffer. Although both previous studies worked in extremely dilute conditions, early phase separation due to local overconcentration most likely occurred, as it is very difficult to spot by eye due its translucent, dispersed nature. The phase being formed may or may not correspond to a dense CaP liquid, the exploration of which is beyond the scope of the present ion association study. In any case, early phase separation leads to higher amounts of bound calcium and overestimation of the ion association constants, especially at higher pH and when assessing $[CaPO_4]^-$ ion pair formation. As shown in Fig. 3 (top), the larger value for $K_3$ from the literature dominates the speciation of CaP ion pairs. The $[CaPO_4]^-$ ion pair is the dominant species from just above pH 8 onwards. Using the values measured here this pH range narrows significantly, leading to $[CaHPO_4]^0$ being dominant up to about pH 10 (Fig. 3, bottom).

Figure 3 makes it clear that the role of $[CaPO_4]^-$ has been overstated, obscuring the importance of $[CaHPO_4]^0$. Especially in the near neutral pH range, which includes physiological pH, $[CaHPO_4]^0$ is much more significant when using the binding constants from this work. Even the effect of a smaller $K_1$ value can be seen with the transition from $[CaH_2PO_4]^+$ to $[CaHPO_4]^0$ as the dominant ion pair being shifted to lower pH. As can be seen in Supplementary Fig. 13, this also affects the total amount of calcium ions bound; using the values from Chughtai et al. nearly 100% of all $Ca^{2+}$ ions are bound starting at a pH of 9[45]. With the updated ion association constants such a high fraction of bound calcium is only achieved above pH 11. Interestingly, even with the large fraction of $PO_4^{3-}$ at such solution conditions (Supplementary Fig. 14), the difference between the two sets of ion association constants can be seen, as around 5% of the $Ca^{2+}$ is predicted to remain free in solution using the constants presented here. Together these

**Table 2 | Thermodynamic quantities for the binding of calcium to the three phosphate ions from experiment (expt) and simulation (sim)**

|                | $\Delta G^0$ (kJ/mol) | $\Delta H^0$ (kJ/mol) | $\Delta S^0$ (J/K/mol) |
|----------------|-----------------------|-----------------------|------------------------|
| $K_1$ (expt)   | −3.7 ± 0.9            | 0.0 ± 77              | 12.5 ± 260             |
| $K_2$ (expt)   | −15.2 ± 0.3          | 1.3 ± 13              | 55.6 ± 43              |
| $K_2$ (sim)    | −14.0 ± 0.5          | 28 ± 5                | 140 ± 15               |
| $K_3$ (expt)   | −26.8 ± 0.5          | 20.8 ± 22             | 160 ± 74               |
| $K_3$ (sim)    | −22.8 ± 0.3          | 15 ± 3                | 128 ± 10               |

The overall reaction is thermodynamically favourable even though the standard enthalpy change is not. The standard enthalpy calculated for the first ion association ($K_1$) is zero – this follows from using the same ion association constant at both temperatures (Supplementary Section 4).

observations indicate that the $[CaHPO_4]^0$ ion pair may be more significant in systems where CaPs are present than has been assumed and modelled in past decades.

In order to theoretically verify the experimentally determined ion association constants, we have performed computer simulations of the ion pairing for both $HPO_4^{2-}$ and $PO_4^{3-}$ with $Ca^{2+}$ ions. As shown previously for the association of calcium ions with carbonate and bicarbonate, simulation results can be significantly improved by the inclusion of polarizability in the underlying force field model since this has a particular influence on the stability of the contact ion pair[54]. In light of this, we have developed an improved, polarisable force field for the simulation of CaP speciation in water based on the AMOEBA model. Full details of the parameterisation and comparison of the performance of this model against experimental data and hydration information from ab initio molecular dynamics can be found in the Methods section. Based on this model, the free energy landscape can be determined for ion pairing as a function of the Ca-P distance and calcium-water coordination number (again see Methods for full details). By appropriate post-processing of this data, it is then possible to compute the standard free energy of ion pairing, leading to values of −14.0 and −22.8 kJ/mol for $[CaHPO_4]^0$ and $[CaPO_4]^-$, respectively. While the thermodynamics of ion pairing remains relatively consistent between the current model and our earlier non-polarisable one for $[CaHPO_4]^0$ (−17.9 kJ/mol), the stability of the calcium-phosphate ion pair is considerably reduced (−35.9 kJ/mol), where the previous values are given in parenthesis. Significantly, the ion pairing free energies computed from the improved model are far more consistent with those determined experimentally in this work (Table 2). While an earlier determination of the free energy of the $[CaPO_4]^-$ ion pair stability using simulation by Tan et al. led to a more exothermic value of −33.1 kJ/mol, it is important to note that this was not corrected to standard conditions and hence should not be directly compared against experiment or our theoretical results[55].

To further examine whether this weaker association for $[CaPO_4]^-$ is plausible, we have also used ab initio quantum mechanical calculations based on solvated clusters to estimate the free energy difference between ion pairing for hydrogen phosphate and phosphate (see Methods section for technical details and discussion of the method). While the absolute ion pairing free energies are challenging to determine accurately, due to the limitations of static cluster approaches in capturing the change in dynamic hydration entropy in particular, the shift between two systems that differ by a single proton should be more reliably computed. Using this approach, the free energy difference between the two contact ion pairs, relative to their separate ions, is estimated to be between 1.9 and 12.2 kJ/mol, depending on the preferred calcium hydration state that is assumed, which compares to a difference of 8.8 kJ/mol from our free energy simulations and

11.6 kJ/mol from our experimental data. Although there is a spread of values associated with the variation in theoretical models, all values are substantially lower than the widely reported difference of 21.4 kJ/mol currently found in the literature.

Additional insight into the ion pairing process can be gained by considering the temperature-dependence of the association. Using our polarisable model, we have determined the variation in the ion pairing free energy over the range from 290 to 350 K (Supplementary Fig. 19). From a linear fit it is then possible to extract the separate enthalpic and entropic contributions (see Table 2). Both the $[CaHPO_4]^0$ and $[CaPO_4]^-$ ion pairs are found to have positive standard enthalpies of formation, while the entropic term is favourable and responsible for driving association. To confirm the theoretical thermodynamics of ion binding experimentally, additional measurements near physiological conditions (37 °C) were performed at pH 8.3 and 11.3 (Supplementary Section 4, Supplementary Figs. 21 and 22). A slight increase in the ion association is seen at this higher temperature, leading to best fit binding values of 4.5, 480 and 69,500 (Supplementary Table 3, overlay of models on data Supplementary Figs. 23–25). The thermodynamic parameters for ion association are calculated using the binding constants from the two temperatures, as summarised in Supplementary Section 4. The results for each ion association are presented in Table 2; the reactions are thermodynamically favourable even though the enthalpy is positive. The change in enthalpy for the first ion association is zero as the same $K_1$ was used for each temperature; based on the small changes in $K_1$ observed by both Gregory et al. and McDowell et al. between 25 and 37 °C we find this approximation appropriate (Supplementary Section 4)[42,51].

Considering the ion association thermodynamics, the enthalpic and entropic balance of $[CaHPO_4]^0$ formation from our titration experiments is reminiscent of the case of calcium carbonate. Here, the slightly endothermic balance does not hinder continuing association toward pre-nucleation cluster formation, while the distinct entropy gain drives this process, due to the release of hydration waters[47]. However, the simulation results indicate a significantly more endothermic balance especially for $[CaHPO_4]^0$, and with it, an even more pronounced entropic contribution to cation-anion association. Still, experiment and simulation are generally in agreement in terms of the standard free energies, even if further refinements to both could narrow the gaps. It should be noted that, experimentally, we have assessed the temperature dependence of the thermodynamics of ion association based on only two closely spaced points, so as to provide a comparison for the computational values. The associated error is thus rather large. The computer simulations, on the other hand, only account for ion pair formation and thereby inherently neglect any processes that might also contribute to the experimentally determined values, where an average is determined. Indeed, additional computer simulations also reveal a strong tendency for hydrogen phosphate anions to form hydrogen-bonded dimers, $(HPO_4)_2^{4-}$, even when separated from $Ca^{2+}$ ions. Given the intrinsic electrostatic repulsion between two anions, this could be considered to be surprising, but can indeed be validated based on free energy determinations using the AMOEBA model, as well as molecular quantum mechanical calculations. While the hydrogen phosphate dimer thus represents a thermodynamically stable minimum in the complex speciation landscape, it is also found to be a kinetic trap during molecular dynamics simulations with several minima existing for the doubly and singly hydrogen bonded states. This made it impossible to reliably determine the association of $[CaHPO_4]^0$ ion pairs beyond dimers toward cluster formation due to the need for collective variables that map not only the distance and relative orientation of the ion pairs, plus the hydration of calcium, but also the hydrogen bonding of the phosphate anions. As a result, determining the further speciation of the system remains a challenge for future work.

## Discussion

The existence of ~1 nm-sized cluster species in CaP solutions has been known for a long time; the clusters are generally considered to be $Ca_9(PO_4)_6$ units, so-called Posner's clusters[35,56–58]. Their speciation and composition were originally proposed based upon XRD radial distribution function analyses of bulk ACP[29,30]. While the arrangement of atoms in Posner's cluster reflects the atomic arrangement of solid HAP, they were also postulated to exist in solution before precipitation occurs, neglecting solution ion association thermodynamics, even though the latter is largely consistent with previously reported ion association constants[36,59]. Aggregation of these clusters was then proposed to directly lead to ACP. However, the current data reveals that actually $[CaHPO_4]^0$ association strongly dominates the CaP speciation in a broad pH range around the neutral regime, casting doubt on the previously proposed dominance of ion associates and clusters with $PO_4^{3-}$, at least until high pH. While $[CaPO_4]^-$ formation still leads to significant binding of calcium at high pH as the phosphate speciation shifts towards $PO_4^{3-}$, this is delayed due to the smaller interaction between the respective ions. The old ion association constants predict near complete binding of calcium ions from solution starting near pH 9, which does not completely occur even above pH 11 using the ion pairing constants presented here. The highlighted role of $HPO_4^{2-}$ in the formation of CaP phases from aqueous solution mirrors developments in the $CaCO_3$ system where the corresponding role of protonated species in nucleation is being uncovered[60]. Moreover, the thermodynamics determined here imply that $[CaHPO_4]^0$ association may proceed toward pre-nucleation cluster formation, potentially with DOLLOP-structural features, where anion-anion interactions may thermodynamically and also kinetically stabilise multi-ion clusters with different Ca/P ratios, and perhaps layered structural motifs[61]. Such species may be involved in the formation of recently described amorphous calcium-hydrogen phosphate and dicalcium phosphate monohydrate phases[10,11]. Additionally the wide range of compositions seen in ACP may be explained by the formation of larger entities, especially if a pH dependence is seen in these structures – this would likely also be relevant for biomineralization pathways to be understood. While the Posner motif may thus not be as stable in solution as previously thought, further computational exploration of the atomistic details of CaP cluster formation based on the revised speciation and thermodynamics will be imperative. However, the combination of needing to examine the revised stability of interactions between calcium and the key phosphate anions, while also considering the tendency of the anions to form competing species in addition, plus proton transfer events, makes for a complex free energy landscape. Consequently, determining the full mechanism for ion association in the aqueous calcium phosphate system, in light of the data reported here, remains a significant challenge for future research.

## Methods

### Titrations

All chemical used were at least analytical grade or titration standard solutions. Phosphate buffers (0.010 M) were prepared using $NaH_2PO_4 \cdot 2H_2O$ (Sigma-Aldrich 75100) and $Na_2HPO_4$ (PanReac 131679.1211) in a ratio near the desired pH and finely adjusted with 1 or 0.1 M HCl and NaOH (Roth K025.1, K024.1, K021.1, K020.1). Calcium chloride stock solution (1 M, VWR Chemicals 190464 K) was diluted to 0.005 M. Ultrapure water (conductivity < 0.05 μS/cm) from a Satorius arium pro device was used exclusively.

Titration was performed using an autotitrator (Metrohm 809 Titrando) equipped with Metrohm 800 Dosino units using 2 mL burettes. pH and $Ca^{2+}$ concentrations were measured using a flat membrane pH electrode (Metrohm, 6.0256.100) and Ca-ISE (Metrohm, 6.0508.110), respectively. The pH electrode was filled with 3 M KCl solution (Merck 1.04817). Calibration of the pH electrode was performed using buffers with pH 4.01, 7.00 and 9.21 (Mettler Toledo 51

302 069, 51 302 047, 51 302 070). Titrations were performed in triplicate.

For the titration, $CaCl_2$ solution was added to 50.00 g well stirred phosphate buffer at 0.01 mL/min; pH was kept constant via counter addition of 0.1 M HCl or NaOH, as needed. The titrations at pH 4.3 were performed at 0.5 mL/min. The titration was kept under nitrogen atmosphere – the gas was passed through a wash bottle to prevent drying out the solution. The added calcium solution was acidified to prevent phase separation at the calcium dosing tip (overview in Supplementary Table 4, further discussion in Supplementary Section 1).

Calibration of the Ca-ISE was carried out identically to the experiments, using NaCl (PanReac 131659.1211) solution with the same pH and ionic strength as the measured phosphate buffer. As calibration is performed in an unbuffered solution, the concentration of NaOH used to maintain the pH was reduced in some cases. This prevented spikes in pH. For the same reason, the HCl in the calcium solution was also replaced with NaCl at a concentration to match the ionic strength. The ionic strengths of these solutions were also adjusted with NaCl to have the same ionic strength as the solutions used in the titration. An overview of the solution concentrations is shown in Supplementary Table 4. For pH 7.3 no difference was seen with acidified vs. non-acidified experiments/calibrations so the non-acidified data was used, also without adjusting the NaOH ionic strength as no effect was seen.

The ionic strength adjustment means that the Ca-ISE reads concentration of $Ca^{2+}$ in the experiment, not activity as when calibrations are done in pure water[48]. The ionic strength was initially calculated using the geochemical software PHREEQC, and the ionic strength varied less than 5.2% from the value calculated for the actual buffers. Calibrations using NaCl solutions with deviation of nearly 10% were tested and no effect was observed on the data (Supplementary Fig. 26).

To ensure that the correct phosphate speciation was used for the ionic strength determination and ion association calculation, the $pK_a$ values for phosphoric acid were experimentally determined in the titration setup, below.

### $pK_a$ titrations

Solutions of 0.010 M phosphoric acid (Carl Roth 2608.1) were titrated with 0.1 and 1 M NaOH stock solutions (Carl Roth K020.1, K021.1) using the same Metrohm titration setup as described for the calcium into phosphate titrations. Various addition rates as defined by the Tiamo 2.5 software and volumes were measured, as summarised in Supplementary Table 5. Methods 5 and 6 had a separate dosing unit adding 0.020 M phosphoric acid with the same addition rate as NaOH to compensate for dilution. For each method experiments were performed in triplicate.

Analysis was done by finding the equivalence points from the derivative of the data and then calculating the half equivalence points. The $pK_a$ values for all 6 methods were nearly identical, averaging to $pK_{a1}$: $2.59 \pm 0.03$ and $pK_{a2}$: $7.00 \pm 0.04$.

To eliminate the possibility of ionic strength effects, additional titrations using Method 1 were performed. The total ionic strength ($I$) was adjusted with NaCl to be within the range seen in the titrations and also extended to higher $I$. Calculation of the phosphate speciation was performed using the $pK_a$ values from the individual experiments, the ionic strength in the titration and the respective $pK_{a2}$ value are shown in Supplementary Fig. 27.

The average value of $pK_{a2}$ in the relevant ionic strength range (~0.015–0.03) does not show an ionic strength dependence. Above $I$ of ~0.05 the $pK_{a2}$ value decreases noticeably. Averaging the $pK_{a2}$ values in the titration relevant region yields a value of $6.98 \pm 0.02$; the corresponding $pK_{a1}$ is $2.58 \pm 0.02$. Although the ionic strength at the $pK_{a1}$ will be slightly lower than that in the calcium-phosphate titrations, the small dependence on $I$ means these values can be used. Additionally, the first $pK_a$ value of phosphate does not have a significant

effect on the binding analysis as it has little influence on the speciation of phosphate ions which show appreciable ion association to calcium. The third $pK_a$ value of phosphoric acid is not accessible via the methods used here so the literature value of 12.32 was used[62].

## Conversion of concentration measurements to activity
Since the ion association/binding constants are defined using activities, the concentrations must be converted. To start the ionic strength ($I$) is calculated as in Eq. 1, below.

$$I = \frac{1}{2} \sum_i z_i^2 \frac{c_i}{c^0} \tag{7}$$

Where ($z_i$) is the ionic charge, ($c_i$) the molar concentration and ($c^0 = 1M$) the standard concentration. This is then used to calculate the activity coefficient for each species, $\gamma_i$, using the Davies Equation[49].

$$\log(\gamma_i) = -0.5085z_i^2\left(\frac{\sqrt{I}}{1+\sqrt{I}} - 0.3I\right) \tag{8}$$

Activity coefficients are charge, not species, dependent; hence $Ca^{2+}$ and $HPO_4^{2-}$ have the same activity coefficient, $\gamma$. The activity coefficients are in the ranges 0.84–0.88, 0.50–0.61 and 0.21–0.33 for singly, doubly, and triply charged species, respectively.

Activity, $a_i$, is then a product of activity coefficient and concentration.

$$a_i = \gamma_i \frac{c_i}{c^0} \tag{9}$$

Activity is generally seen as unit-less; this is done by dividing $c_i$ by $c^0 = 1\,M$. Converting the concentration of free calcium into activity, the titration data transforms into Supplementary Figs. 5 and 6. Since the value for $\gamma_{Ca^{2+}}$ varies only a few percent for each pH, the plot looks nearly identical, with the y-axis reduced by ~50%.

## Ion pairing models
**Direct calculation: experimental value-based model.** Assuming there is only 1:1 ion association, constants, $K_i$, for the interaction between calcium and the various phosphate ions are defined based on the following reactions:

$$Ca^{2+} + H_2PO_4^- \rightleftharpoons [CaH_2PO_4]^+ \tag{10}$$

$$Ca^{2+} + HPO_4^{2-} \rightleftharpoons [CaHPO_4]^0 \tag{11}$$

$$Ca^{2+} + PO_4^{3-} \rightleftharpoons [CaPO_4]^- \tag{12}$$

The constants are then defined using the law of mass action:

$$K_1 = \frac{a_{[CaH_2PO_4]^+}}{a_{Ca_{free}^{2+}} a_{H_2PO_4^-}} \tag{13}$$

$$K_2 = \frac{a_{[CaHPO_4]^0}}{a_{Ca_{free}^{2+}} a_{HPO_4}} \tag{14}$$

$$K_3 = \frac{a_{[CaPO_4]^-}}{a_{Ca_{free}^{2+}} a_{PO_4^{3-}}} \tag{15}$$

Where $a_i$ denotes the activity of a species. Additionally, there are several mass balances for the system:

$$n_{Ca_{added}^{2+}} = n_{Ca_{free}^{2+}} + n_{Ca_{bound}^{2+}} \tag{16}$$

Where $n_i$ describes the moles of a species present in the titration. According to the assumed binding stoichiometry above, the amount of bound calcium is directly proportional to the total number of ion pairs in the system.

$$n_{Ca_{bound}^{2+}} = n_{[CaH_2PO_4]^+} + n_{[CaHPO_4]^0} + n_{[CaPO_4]^-} \tag{17}$$

Using the first ion pair ($[CaH_2PO_4]^+$) as an example, the molar amount of an ion pair can be related to the binding constant. The designation $c_i$ is used to represent concentrations.

$$a_{[CaH_2PO_4]^+} = K_1 a_{Ca_{free}^{2+}} a_{H_2PO_4^-} \tag{18}$$

$$c_{[CaH_2PO_4]^+} = \frac{a_{[CaH_2PO_4]^+}}{\gamma_{[CaH_2PO_4]^+}} = \frac{K_1 a_{Ca_{free}^{2+}} a_{H_2PO_4^-}}{\gamma_{[CaH_2PO_4]^+}} \tag{19}$$

$$n_{[CaH_2PO_4]^+} = \left(c_{[CaH_2PO_4]^+}\right) V_t = \left(\frac{K_1 a_{Ca_{free}^{2+}} a_{H_2PO_4^-}}{\gamma_{[CaH_2PO_4]^+}}\right) V_t \tag{20}$$

Where $V_t$ is the total volume of the solution.

The activity of the phosphate ions is also calculated from a mole balance. Assuming 1:1 binding the amount of each ion can be calculated from the pH, solution volume and experimentally determined amount of bound $Ca^{2+}$.

$$n_{P,free} = n_{P,total} - n_{Ca_{bound,expt}^{2+}} \tag{21}$$

Using $[H_2PO_4]^-$ as an example:

$$a_{H_2PO_4^-} = \gamma_{H_2PO_4^-}\left(\frac{f_{H_2PO_4^-} n_{P,free}}{V_t}\right) \tag{22}$$

Where $f_{H_2PO_4^-}$ is the fractional abundance of $H_2PO_4^-$ in the solution based on pH.

Repeating the procedure for the remaining two ion pairs and then substituting into Eqs. 16 and 17, the amount of free calcium can be calculated.

$$n_{Ca_{free}^{2+}} = n_{Ca_{added}^{2+}} - \left(\left(\frac{K_1 a_{Ca_{free}^{2+}} a_{H_2PO_4^-}}{\gamma_{[CaH_2PO_4]^+}}\right) V_t + \left(\frac{K_2 a_{Ca_{free}^{2+}} a_{HPO_4^{2-}}}{\gamma_{[CaHPO_4]^0}}\right) V_t \right.$$
$$\left. + \left(\frac{K_3 a_{Ca_{free}^{2+}} a_{PO_4^{3-}}}{\gamma_{[CaPO_4]^-}}\right) V_t\right) \tag{23}$$

This can then be simplified:

$$n_{Ca_{free}^{2+}} = n_{Ca_{added}^{2+}} - a_{Ca_{free}^{2+}} V_t\left(\left(\frac{K_1 a_{H_2PO_4^-}}{\gamma_{[CaH_2PO_4]^+}}\right)\right.$$
$$\left. + \left(\frac{K_2 a_{HPO_4^{2-}}}{\gamma_{[CaHPO_4]^0}}\right) + \left(\frac{K_3 a_{PO_4^{3-}}}{\gamma_{[CaPO_4]^-}}\right)\right) \tag{24}$$

Finally, the mole balance is converted into the relevant activity:

$$a_{Ca_{free}^{2+}} = \frac{n_{Ca_{free}^{2+}}}{V_t}\gamma_{Ca^{2+}} = \frac{\gamma_{Ca^{2+}}}{V_t}\left(n_{Ca_{added}^{2+}} - a_{Ca_{free}^{2+}}V_t\left(\left(\frac{K_1 a_{H_2PO_4^-}}{\gamma_{[CaH_2PO_4]^+}}\right)\right.\right.$$
$$\left.\left. + \left(\frac{K_2 a_{HPO_4^{2-}}}{\gamma_{[CaHPO_4]^0}}\right) + \left(\frac{K_3 a_{PO_4^{3-}}}{\gamma_{[CaPO_4]^-}}\right)\right)\right) \qquad (25)$$

Using a mole balance and converting to activity at the very end elegantly avoids any issues with performing manipulations on activities. The molar amount of any species is determined by the initial conditions and values (either measurement or dosing) readily available from the experimental setup. It needs to be noted that the model includes the activity coefficient of a neutral species ($\gamma_{[CaHPO_4]^0}$). In dilute conditions the activity of neutral species is taken to be unity since these molecules will be well separated in solution and not affected by electrostatic forces, as charged species would be in the same solution[63]. Formally assigning $\gamma_{neutral} = 1$ and subsequently $a_{neutral} = c_{neutral}$ also has precedent in earlier works looking concerning ion association for calcium phosphate[41]. Mathematically this also follows from the Davies Equation when inserting a charge of 0.

### Predictive calculation: pure law of mass action model

The law of mass action can also be used to predict the free calcium activity based only upon the added amounts of reagents, the phosphate speciation and activity coefficients. This avoids using experimentally determined concentrations in the calculation.

Using the first ion pair ($[CaH_2PO_4]^+$) as an example, Eq. (4) can be rewritten:

$$K_1 = \frac{\gamma_{[CaH_2PO_4]^+} c_{[CaH_2PO_4]^+}}{\gamma_{Ca_{free}^{2+}} c_{Ca_{free}^{2+}} \gamma_{H_2PO_4^-} c_{H_2PO_4^-}} \qquad (26)$$

Converting concentrations to moles and rearranging:

$$K_1 = \frac{\gamma_{[CaH_2PO_4]^+}}{\gamma_{Ca_{free}^{2+}} \gamma_{H_2PO_4^-}} \frac{1/V_t}{1/V_t 1/V_t} \frac{n_{[CaH_2PO_4]^+}}{n_{Ca_{free}^{2+}} n_{H_2PO_4^-}} \qquad (27)$$

Using mole balances from Eqs. (16) and (17) for calcium and similar for the phosphates:

$$K_1 = \frac{\gamma_{[CaH_2PO_4]^+} V_t}{\gamma_{Ca_{free}^{2+}} \gamma_{H_2PO_4^-}} \frac{n_{[CaH_2PO_4]^+}}{\left(n_{Ca_{added}^{2+}} - n_{[CaH_2PO_4]^+} - n_{[CaHPO_4]^0} - n_{[CaPO_4]^-}\right)\left(f_{H_2PO_4^-}\left(n_{PO_{4\,initial}} - n_{[CaH_2PO_4]^+} - n_{[CaHPO_4]^0} - n_{[CaPO_4]^-}\right)\right)} \qquad (28)$$

This calculation is set up similarly for the remaining two ion pairs. The system of equations is then solved using a symbolic algebra solver[64].

### Computational details

**Ab initio molecular dynamics.** Ab initio molecular dynamics simulations of both the phosphate and hydrogen phosphate ions in water were performed to provide information regarding the structure of the local hydration shell surrounding the species. All calculations were performed using the CP2K code using the Gaussian-augmented planewave approach, as embodied in the Quickstep methodology[65,66]. Norm-conserving pseudopotentials of the GTH form were employed in combination with the TZ2P basis set for the valence electrons[67]. Calculations were performed using the PBE functional in combination with D3 dispersion corrections[68–70]. Self-consistency was achieved using the orbital transformation algorithm with full kinetic preconditioning[71]. Simulations used a cubic box of length 1.43 nm

containing the relevant ion and 99 water molecules starting from a configuration equilibrated with a classical force field. Molecular dynamics was then performed with a timestep of 0.5 fs and the mass of hydrogen was set equal to that of deuterium. An elevated temperature of 330 K was used with the CSVR thermostat to partially offset the systematic over-structuring of water, as has been previously proposed[72,73].

### AMOEBA

All the classical molecular dynamics simulations presented in this work have been run with the OpenMM toolkit on GPUs using mixed precision[74]. The AMOEBA force field developed in this work was used in conjunction with the current AMOEBA 03 water model[75,76]. The atomic multipoles were determined by fitting the electrostatic potential obtained from MP2/6-311 G(d,p) density functional theory calculations using the distributed multipole analysis method[77]. It is worth mentioning here that the direct application of this method would produce an asymmetric multipole for $HPO_4^{2-}$, which would result in different energies for the three equivalent orientations of the OH bond as a function of rotation about its torsional angle. Hence, we have manually set to zero all dipole and quadrupole terms for the phosphorous atom and the three equivalent oxygen atoms to enforce the correct symmetry with respect to the torsional energy profile. All simulations were run with fully flexible molecules using a 1 fs timestep in 3D periodic boundary conditions and the temperature was controlled using a Langevin thermostat. The simulation cells were initially equilibrated at the desired temperature and 1 atm using the isotropic Monte Carlo barostat, and all production runs were performed in the NVT ensemble using the average equilibrated cell from the NPT simulations. The electrostatics were computed with the Particle Mesh Ewald (PME) algorithm with a precision of 1e-5 and the mutual polarisation scheme with a 1e-5 convergence threshold.

A 1 ns MD run for each of the isolated anions in water was used to characterise the structure of their hydration shell and compare against DFT calculations performed in this work. The anion-water radial pair distribution functions and the 3D water density were then computed to validate the force field parameterisation. The hydration free energy of the ions was computed using free energy perturbation and the Bennett acceptance ratio methods[78,79]. The calculations were run with one ion immersed in a cubic box with a side length of approximately 2.5 nm, with 50 equally spaced stages each for the perturbation of both the van der Waals and electrostatic interactions. In the case of $HPO_4^{2-}$, one extra free energy perturbation calculation was carried out in vacuum to account for the intramolecular Coulomb interactions. A finite size correction was also applied to allow for the creation of a charge in a 3D periodic system[80,81]. In addition to the AMOEBA parameters as fitted to the hydration structure and free energy, the interactions between $Ca^{2+}$ and the oxygens of the phosphate and hydrogen phosphate anions were augmented by a Buckingham potential. As previously highlighted for the case of calcium-carbonate/bicarbonate interactions, the addition of a small amount of extra repulsion can be used to fit the solubility of a relevant mineral phase. In the present work, the bulk dissolution free energies of beta tricalcium phosphate and monetite were used to fit the Buckingham A parameter for $Ca^{2+}$ interacting with $PO_4^{3-}$ and $HPO_4^{2-}$, respectively, while keeping the rho value fixed at 0.03 nm[54]. This resulted in similar Buckingham A values of 243.5 and 160 eV for phosphate and hydrogen phosphate, again respectively. All bulk fitting and optimisations were performed with an

in-house modified version of GULP coupled to Tinker[82]. A repulsive Buckingham potential between the water oxygen and the hydrogen of $HPO_4^{2-}$ with an A parameter of 100 eV and the same rho value was added to improve the hydrogen phosphate by water radial pair distribution function. The ion pairing free energy of the two phosphate anions considered with $Ca^{2+}$ was computed using the multiple-walker well-tempered metadynamics method as implemented in OpenMM[83–85]. A larger box of approximately 3 nm was used to allow for the alignment of the free energy with the analytic long-range limiting solution and the calculation of the ion association constant, as discussed in Aufort et al. [86]. The simulations were run using 8 walkers with a bias factor of 5, a deposition rate of 1 ps and an initial Gaussian height of 2.5 kJ/mol. The simulations were run for an aggregate time of at least 250 ns. Two collective variables (CVs) were chosen, the Ca-P distance and the $Ca^{2+}$ by water coordination number, and the width of the Gaussians was set to 0.01 nm and 0.1 for the two CVs, respectively. The Ca by water coordination number was computed as;

$$\sum_i \frac{1}{2} erfc\left(\frac{r_i - 0.33}{0.045}\right) \tag{29}$$

where $r_i$ are the distances between the Ca ion and all the water oxygen atoms in nm. The distance CV was limited by inclusion of a harmonic upper wall at 1.2 nm, but the calculation of the ion association constant was computed by extrapolating the 1D pairing free energy up to the Bjerrum length based on the alignment with the asymptotic limit for the screened Coulomb interaction and entropy. This approach can be contrasted with the earlier calculation of Ca-PO₄ ion pairing of Tan et al. who used a non-polarisable force field with a single distance collective variable, truncated at 0.8 nm, and only 10 ns of data collection, without correction to the standard state[55]. Therefore, we contend that the present approach should be better converged and more accurate. The ion association constant was computed at 7 temperatures in the 290–350 K range in triplicate (i.e. each simulation was performed three times from different initial conditions to obtain estimates of the statistical uncertainty) to extract the enthalpy and entropy of association via a linear fit.

See Supplementary Section 3 for force field parameters and Supplementary Figs. 15–20 for results of the AMEOBA simulations.

## Molecular quantum mechanical calculations

To complement the ion pairing free energies from classical molecular dynamics, calculations were also performed using molecular quantum mechanical calculations. Here it is important to note that determining absolute binding energies for species in water is challenging because of the limitations of the description of the hydration environment. Therefore, the emphasis was primarily on trying to determine the difference in binding free energy between the $[CaPO_4]^-$ and $[CaHPO_4]^0$ ion pairs. It is well known that the use of continuum solvation models alone can fail to capture the thermodynamics of hydration, especially when hydrogen bonding is important, as is the case for the phosphate anions. Hence in the present study we therefore use microsolvation of $Ca^{2+}$ by either 6 or 7 water molecules, given the well-known variability in the hydration sphere of this ion, while the phosphate anions are both surrounded by 12 waters[87]. In all cases the SMD solvation model is then used to supplement the explicit first hydration shell[88]. Two steps have been taken to try to minimise the uncertainty arising from the use of a static hydration environment and the known limitations of this approach. Firstly, for the ion pairs the number of explicit waters is set to 18 or 19 for 6- or 7-fold water coordination of $Ca^{2+}$, respectively, such that the number of explicit water molecules is conserved during the reaction. Secondly, the optimised configuration of water molecules around the phosphate ion (or corresponding ion pair) was used as the starting structure for the hydrogen phosphate case, with the addition of the extra proton. This should minimise any variation that might

come from different choices of configuration for the explicit water shell that could otherwise mask a small free energy change.

All molecular quantum mechanical calculations were performed at the ωB97X-D3/ma-def2-QZVPP level of theory using the ORCA code, version 5.0.3[89,90]. Following optimisation, both in the presence and absence of the SMD continuum solvation model using the parameters appropriate to water, the binding energies of the ion pairs were computed from the SMD values, but with thermal and zero point contributions from the vibrational frequencies computed in the gas phase, along with a correction for the standard state from 1 atm to 1 M, as appropriate to the solution state.

The absolute free energies of ion pairing for $[CaHPO_4]^0$ and $[CaPO_4]^-$ were computed using the above protocol. For the $[CaPO_4]^-$ contact ion pair (CIP), this system was found to exhibit both monodentate and bidentate coordination of the calcium ion to phosphate. When using 18 water molecules, while both coordination states are stable in the presence of the continuum solvation model, the gas phase optimisation to compute the vibrational frequency contributions always led to bidentate coordination. Addition of an extra water molecule was found to stabilise the monodentate state in the gas phase. Regardless of the number of explicit waters included, the bidentate state was found to be more stable by 1.7 or 12.5 kJ/mol, for 18 or 19 waters, respectively. As a consequence of the lack of stability of the monodentate state in the gas phase for 18 waters, it was only possible to fully compute the ion pairing free energy of the bidentate (bi) and monodentate (mono) states for the 19 water case.

Based on 18 water molecules in total, the ion pairing free energies of $[CaPO_4]^-$ and $[CaHPO_4]^0$ were determined to be −43.5 and −41.6 kJ/mol, respectively. In comparison, for the 19 water case (which allows Ca to adopt 7-fold water coordination in solution), the corresponding values are −36.4 (bi)/−23.9 (mono) and −24.2/−20.0 kJ/mol, again respectively. Here two values are given for the $[CaHPO_4]^0$ ion pair, both based on monodentate Ca-HPO₄ binding, but with different initial configurations of water. The foregoing values illustrate the challenge of computing the absolute values accurately, even when using one of the more reliable forms of density functional theory, due to the difficulties of capturing the dynamic and variable nature of the explicit hydration shell structure, and the associated entropy when using a molecular static approach. For this reason, the use of free energy methods based on molecular dynamics (as performed in the previous section) is more likely to be accurate for the absolute values, provided the force field description of water is reasonable. Although the ion pairing free energy for calcium-phosphate is close to several previous literature values, this is coincidental. Repeating the calculations for 18 waters with the same functional, but a def2-TZVP basis set shifts the values to −54.4 and −50.9 kJ/mol when given in the same order as previously. This illustrates that the absolute value is easily shifted by choices within method, but the difference in values remains consistent and small. It should also be noted that the difference in ion pairing free energies for 18 waters is 8.6 kJ/mol based only on the internal energy (including SMD solvation) but is reduced by the zero point and thermal vibration contributions. Focusing on the more reliable relative free energies of the ion pairs, if we take the difference between the most stable configurations in each case this gives values of 12.2 and 1.9 kJ/mol for 19 and 18 waters, respectively. While there is considerable variability depending on the hydration model, the key point is that both values are considerably lower than current experimental literature estimates, which would suggest a difference of ~21.4 kJ/mol and are more consistent with the values determined in this work (11.6 kJ/mol).

Given that the AMOEBA results suggest that the solvent-shared ion pair state (SSHIP) is more stable than the contact ion pair, we also attempted to consider such configurations using molecular quantum mechanical calculations. Here configurations were generated where a single water molecule bridged between the two ions. Applying the

same procedure to the SSHIP for [CaPO$_4$]$^-$ was found not to be possible since the gas phase optimisation always led to formation of a contact ion pair. Comparing the internal energy differences alone would indicate that the SSHIP for this ion pair is less stable than the CIP by 44−57 kJ/mol. For [CaHPO$_4$]$^0$, the absolute free energies of binding for the SSHIP state were all close to the order of thermal energy (−2.8 to +0.5 kJ/mol), again making it less stable than the CIP. Rather than suggesting that the AMOEBA calculations are in error, these results are more likely to indicate that this approach is not reliable for such configurations in water where there are many configurational possibilities, and the dynamics of water and entropy are even more important than for the contact state.

For Ca$^{2+}$, the calculation of the hydration free energy was performed using both the monomer and cluster methods of Bryantsev et al. at the level of theory given above, including zero point energy and thermal corrections for the gas phase binding, and using a cluster of 12 water molecules in the latter approach[91]. This results in hydration free energies for this cation of either −1518 (−1523) or −1551 (−1558) kJ/mol for the monomer and cluster methods, respectively, where the values are given for 6-fold coordination of Ca$^{2+}$ by water (or 7-fold in parenthesis). These values lie within the bounds of those from Marcus and the earlier quantum mechanical values of Florián and Warshel (see Supplementary Table 2), and indicate that the absolute hydration free energy is sensitive to the details of the reference state of the water molecules (i.e., a single molecule or a cluster of waters) included alongside the continuum model[92]. The hydration free energy of Ca$^{2+}$ is also systematically lower (by 5−7 kJ/mol) for 7-fold coordination by water, which is consistent with previous reports of an average coordination number closer to 7 than 6, including from the AMOEBA-based model[54]. However, the corresponding monomer and cluster method hydration free energies for Ca$^{2+}$ with 8-fold coordination are −1527 and −1565 kJ/mol, respectively, showing a further systematic decrease. In the monomer method, the experimental hydration free energy of water was used (−26.4 kJ/mol). However, if the SMD hydration free energy for a single water at the current level of theory is used instead (−31.1 kJ/mol), then the hydration free energies of Ca$^{2+}$ for 6, 7 and 8-fold coordination by water are −1490.8, −1490.7 and −1489.6 kJ/mol, respectively. This suggests that the majority of the free energy difference between hydration numbers is a consequence of the reference free energy for the water molecule in the liquid phase. For the case of the multiply charged phosphate anions, the direct calculation of the hydration free energy is not especially meaningful since the species are formally unstable in the gas phase, with the second (or third) excess electrons only being bound due to the finite extent of the Gaussian basis set. Therefore, the values are particularly sensitive to the calculation details and so not given here.

### Ion association beyond ion pairs

A possible consideration when studying the speciation of calcium and phosphate anions in aqueous solution is whether formation of entities beyond the ion pair might influence the state of the system and if so at what concentration. Consequently, attempts were made to compute the free energy for the dimerization of [CaHPO$_4$] ion pairs in water using AMOEBA-based simulations. However, this revealed a strong tendency for hydrogen phosphate anions to form hydrogen-bonded dimers, (HPO$_4$)$_2^{4-}$, even when separated from Ca$^{2+}$. Given the intrinsic electrostatic repulsion between two anions, this could be considered to be surprising. Therefore, it was necessary to validate whether this behaviour is correct or an artefact of the force field model. To examine whether the strength of binding is realistic, molecular quantum mechanical calculations were performed for the same process using the same level of theory as described earlier (i.e. ωB97X-D3/ma-def2-QZVPP with SMD solvation). For binding of two HPO$_4^{2-}$ anions without an explicit solvation shell to form a doubly-hydrogen bonded dimer this gives a binding free energy of −51.4 kJ/mol. Given the importance of the relative strength of hydrogen bonding between the anions to that of the anions with water, the calculation was also repeated for the reaction:

$$2(H_2O - HPO_4^{2-})(aq) \rightarrow (HPO_4)_2^{4-}(aq) + (H_2O)_2(aq) \quad (30)$$

The free energy for this process was computed to be −23.5 kJ/mol. While it is hard to determine the exact binding free energy in water, the QM values for the dimerization free energy confirm that binding between hydrogen phosphate anions is indeed favourable. Further support for the stability of such hydrogen bonded dimers comes from the crystal structures of other materials. While the majority of structurally characterised dimers consist of dihydrogen phosphate pairs (or even oligomers) with the same motif, there are occasional examples of hydrogen phosphate dimers such as for SnHPO$_4$[93–95].

In addition to the hydrogen phosphate dimer being a stable minimum, it is also found to be a kinetic trap during simulations with several minima for the doubly and singly hydrogen bonded states. This made it impossible to reliably determine the association of [CaHPO$_4$]$^0$ ion pairs at present due to the need for collective variables that map not only the distance and relative orientation of the ion pairs, plus the hydration of calcium, but also the hydrogen bonding of the phosphate anions. As a result, determining the further speciation of the system remains a challenge for future work.

## Data availability

All data generated in this study have been deposited on Zenodo (https://doi.org/10.5281/zenodo.10521019). Data are also available from the corresponding authors upon request.

## Code availability

All simulations have been performed with publicly available software. A few custom python/shell scripts for analysis are provided with the data at https://doi.org/10.5281/zenodo.10521019.

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

## Acknowledgements

This research was supported by Curtin University and the Australian Research Council (FL180100087). The Pawsey Supercomputing Centre and the Australian National Computational Infrastructure are also acknowledged for the provision of computing time through the NCMAS and Pawsey Partners merit allocation schemes. We would like to thank Maxim Gindele for the helpful discussions and Stella Kittel for the generous support in lab.

## Author contributions

D.P.McD. performed all titration experiments, sample preparations, and wrote the original paper draft. J.D.G and P.R. performed the simulations and wrote the corresponding sections. D.G. supervised the experiments and developed the project idea. The paper was written through the contributions of all authors.

## Funding

## Competing interests

The authors declare no competing interests.
