## [Peer Review File · Nature Communications]

Reviewers' Comments:

Reviewer #1:

Remarks to the Author:

In this manuscript the authors combine titration with computational simulation to revisit the ion association constant for calcium phosphate system in aqueous solution. This is a well-designed piece of work that follows up the phosphate mineralization area that has been fairly researched both in the same group and other researchers. Notably, this paper tried to bring new insights into ion association before the nucleation or phase separation for the calcium phosphate minerals. The main conclusion drawn in this study, which suggests smaller values in K1 and K3 would result in more significant existence of [CaHPO₄] complex across larger pH range by two more pH units, are supported by the data presented in both the manuscript and supporting information. Overall, this work may receive some interest given the critical calcium phosphate mineralization for both material and medical applications. However, the levels of insights and novelty are not strong enough for publication in the target journal; therefore, it is recommended this manuscript be submitted to more specialized journal (JPC, Cryst Growth & Design etc) once the authors address the following comments:

1. The error bars for K1, K2 and K3 are missing in Table 1. The error bars have to be added based on the repeated measurements claimed by authors. Similar issue for data in Tables 2, S4, S5, S6 and S7 should be fixed.
2. Based on the curves in Fig. S13, the results of this manuscript are more close to those from Chughtai et al at pH 2, pH 7 and pH 12 compared to the other pH values in between. However, the deviation between the measurement and the calculation using constants from Chughtai et al monotonically (showed in Figure S12) increases as pH increase from 4.3 to 7.3 and finally to 11.3. How could author figure out this discrepancy?
3. Although titration has been used to study the kinetic process of mineralization such as calcium carbonate and calcium phosphate, this technique has a lot of potential problems due to the sluggish addition of one ion into another as well as the high local supersaturation created during addition. The mixed ions system keeps evolving during the addition of new ions, which mess up the solution composition especially for metastable states in high pH for calcium phosphate. The state of mixed solution by titration is deviated from the "ideal state" by mixing ions immediately. That is the reason why the results are also depending on the titration speed. Authors need to think more carefully of such issues.

Reviewer #2:

Remarks to the Author:

Basically, this is a very interesting work, which may provide significant insights into the association process of calcium and phosphate ions in aqueous environment. This work can bring up some new understandings to this fundamental physical chemical issue. Generally, I recommend this manuscript to be accepted for publication in NC. Overall, the computational protocols were well designed and described. Several minor issues are listed below.

1. Please provide initial structures for the association free energy calculations.
2. The reason to choose only six water molecules as the first shell to calcium ions, since many previous work showed 6-8 water might occur around the calcium as the first shell. If possible, more calculations may be conducted for comparison.
3. Some related literatures should be cited for comparison, e.g., Tan et al., Cryst. Growth Des. 2020, 20, 4561.
4. Forms for some references should be carefully checked and corrected. Moreover, Ref21 lacks page number.
5. Line 252, "... have a developed an improved..." should be "... have developed an improved..."

Point by point replies (NCOMS-23-40406-T)

Our replies appear indented and italicised. Corresponding changes are highlighted in the revised manuscript in yellow.

We will publish the source data on Zenodo in case our work is accepted for publication in Nature Communications. Currently, source data can be accessed at:

https://curtin-my.sharepoint.com/:f/a/personal/237454k_curtin_edu_au/EuN54g1w3hFGhzSvTOe7Bv0BLBNFUNB8EH8KMhXfYMFz0Q?e=1Lfeas

Reviewer 1:

In this manuscript the authors combine titration with computational simulation to revisit the ion association constant for calcium phosphate system in aqueous solution. This is a well-designed piece of work that follows up the phosphate mineralization area that has been fairly researched both in the same group and other researchers. Notably, this paper tried to bring new insights into ion association before the nucleation or phase separation for the calcium phosphate minerals. The main conclusion drawn in this study, which suggests smaller values in K1 and K3 would result in more significant existence of [CaHPO₄] complex across larger pH range by two more pH units, are supported by the data presented in both the manuscript and supporting information. Overall, this work may receive some interest given the critical calcium phosphate mineralization for both material and medical applications. However, the levels of insights and novelty are not strong enough for publication in the target journal; therefore, it is recommended this manuscript be submitted to more specialized journal (JPC, Cryst Growth & Design etc) once the authors address the following comments:

We appreciate the Reviewer's very positive assessment of our work; however, we beg to strongly differ with their opinion regarding the general importance of our findings, and the suitability of our manuscript for publication in Nature Communications. Without doubt, the re-determined ion association constants represent a paradigm change in the calcium phosphate field. Not only do they affect (non-classical) cluster-based nucleation theories, but also the speciation and thermodynamics of solutions, amorphous and crystalline intermediates as well as corresponding pathways to final stable phases, and with it, their kinetics. Due to the vast importance of calcium phosphates in chemistry, (bio)medical research and also geological contexts, we strongly feel that our findings are of great importance to a broad scientific audience, rendering publication in a high impact journal literally imperative.

1. The error bars for K1, K2 and K3 are missing in Table 1. The error bars have to be added based on the repeated measurements claimed by authors. Similar issue for data in Tables 2, S4, S5, S6 and S7 should be fixed.

We appreciate the Reviewer's comment on the errors associated with the measurements. While we attempted to provide an overview of the possible spread of ion association constants with the "Limit" cases to proactively cover this point, we can understand where the presentation may not be completely clear for the reader. The revised manuscript has been updated with more traditional error values in addition to the limit cases. We want to highlight

that the multiple measurements, as can be seen in Figure 2, Supplementary Figures 3 and 4, have most certainly been carried out and show good agreement as demonstrated by the small error bars (the standard deviation of the measurements is generally under 5% of the value, quite often around 2%). The error in the ion pairing constants does not come from uncertainties in replicate measurements, which is much lower, but rather the global fitting of the models across all ten separate pH values studied.

2. Based on the curves in Fig. S13, the results of this manuscript are more close to those from Chughtai et al at pH 2, pH 7 and pH 12 compared to the other pH values in between. However, the deviation between the measurement and the calculation using constants from Chughtai et al monotonically (showed in Figure S12) increases as pH increase from 4.3 to 7.3 and finally to 11.3. How could author figure out this discrepancy?

This is a great comment by the reviewer, showing how the consequences of the smaller ion association constants need to be thoroughly discussed from all angles to avoid confusion. Indeed, we find that the figures mentioned complement each other, and along with Figure 3 and the phosphate ion speciation (Supplementary Figure 14, added to SI), demonstrate how the ion pair distribution shifts with the updated ion association constants. Indeed, as shown in Supplementary Figure 12, the deviation increases monotonically with pH; this is directly reflected in Figure 3, where the $[CaPO_4]$ ion pair dominates the speciation starting just after pH 8, where the fraction of bound calcium, as per Supplementary Figure 13, also increases steeply. Using the values from Chughtai et al., the total amount of Ca^{2+} ions bound is essentially 100% from pH 9 onwards. Using the ion association constants from this work, this is delayed until nearly pH 11. From the phosphate speciation (Supplementary Figure 14), it can be seen that the PO_4^{3-} ion begins to be present in significant quantities just above pH 10, directly in this pH range. As the concentration of the phosphate ion increases with pH, the fraction of bound calcium in Supplementary Figure 13 must also increase, even when using the smaller K_3 value. The deviation from the data using the "old" ion association constants is exactly why this increase occurs at higher pH with the updated constants. The difference in magnitude between the two K_3 values, can still be seen at pH 12, with Chughtai et al.'s values predicting essentially 100% binding of calcium while the values from this work suggest around 95% of the Ca^{2+} in the system will be bound.

The closeness of the two curves in Supplementary Figure 13 at pH 2 and 7 can be explained by similar examination. In these pH ranges, the PO_4^{3-} ion and consequently the $[CaPO_4]$ do not play a significant role (as can be seen in the phosphate speciation (Supplementary Figure 14), and the ion pair distribution (Figure 3)). The values for K_1 and K_2 calculated in this work are fairly comparable to those in the literature, indeed the K_2 values are nearly identical, leading to the very small gap around pH 7. The first ion association constant from the literature is slightly larger than the one from this work leading to a slightly more binding occurring at low pH, while the closeness at pH 2 simply results from the limited amount of ion association possible in such conditions.

Comments to help the reader interpret the graphs have been added at the relevant areas in the text.

3. Although titration has been used to study the kinetic process of mineralization such as calcium carbonate and calcium phosphate, this technique has a lot of potential problems due to the sluggish addition of one ion into another as well as the high local supersaturation created during addition. The mixed ions system keeps evolving during the addition of new ions, which mess up the solution composition especially for metastable states in high pH for calcium phosphate. The state of mixed solution by titration is deviated from the “ideal state” by mixing ions immediately. That is the reason why the results are also depending on the titration speed. Authors need to think more carefully of such issues.

While we appreciate the Reviewer’s feedback that more clarification is needed, we are puzzled by the comment on kinetic effects in the titration, when the work presented here focuses on the thermodynamics of ion association. We suspect that they may be making reference to our recent paper on the effect of titration addition rate on calcium phosphate nucleation, where it was shown that kinetics do play a role, however, after nucleation.¹ The reviewer will be pleased to see that such possible effects were in fact given great and thorough consideration. Indeed, it can be seen in the above-mentioned paper that the slope characterizing the ion association in the pre-nucleation regime is independent of the addition rate used, i.e., the slopes of the pre-nucleation regime remain constant with addition rate; kinetics only become important after crystal formation has already begun, which is not the subject of our current manuscript. Similar rate independence on the pre-nucleation slopes has also been seen in the calcium carbonate system.² The ion association studied in the current work exists precisely in this thermodynamically controlled pre-nucleation region. An additional figure has been included in the Supplementary Information to demonstrate this (Supplementary Figure 7). Here it can be seen at two pH values that increasing the addition rate by a factor of 10, even 100, does not change the slope of the measured free Ca^{2+} curves; this directly shows that the ion association is not affected by kinetics and does not depend on the addition speed, but is purely under thermodynamic control. This work specifically looks at this region in the titration.

The Reviewer also mentions a supposedly “sluggish addition” of ions causing local supersaturation. Again, they will be pleased to see that this is not the case. When stopping the addition of Ca^{2+} ions into the phosphate buffer, the measured potential (and therefore the concentration of calcium ions), remains constant until the addition is restarted, as depicted in the added Supplementary Figure 8. Altogether these measurements indicate that the solution is in equilibrium immediately on addition of calcium; the measured values truly represent the entire system. Any local supersaturation would lead to an increase in free Ca^{2+} concentration while addition was stopped, which is clearly not the case here. The solution composition remains constant throughout the pause – overall a collection of linear segments is recorded, simply separated by the waiting time. Converting the time axis in Supplementary Figure 8 to the concentration of added calcium would result in a straight line as presented in the other figures. Furthermore, we would like to highlight the significant effort expended to develop a reliable titration method for this system, with the acidification especially introduced to guarantee that true system-wide behaviour is captured when measuring in the solution.

Reviewer 2:

Basically, this is a very interesting work, which may provide significant insights into the association process of calcium and phosphate ions in aqueous environment. This work can bring up some new understandings to this fundamental physical chemical issue. Generally, I recommend this manuscript to be accepted for publication in NC. Overall, the computational protocols were well designed and described. Several minor issues are listed below.

We thank the reviewer for their very positive assessment of our work and have made revisions to address all of the minor points raised, as listed below.

1. Please provide initial structures for the association free energy calculations.

Input files for the association free energy calculations, including the initial configurations, will be available at Zenodo (DOI: 10.5281/zenodo.10521019). Currently, the data can be accessed at:

https://curtin-my.sharepoint.com/:f/q/personal/237454k_curtin_edu_au/EuN54q1w3hFGhzSyTOe7BvOBLBNFUNB8EH8KMhXfYMFz0Q?e=1Lfeqs

2. The reason to choose only six water molecules as the first shell to calcium ions, since many previous work showed 6-8 water might occur around the calcium as the first shell. If possible, more calculations may be conducted for comparison.

The referee is quite correct, in that the calcium ion is well known to exhibit variable water coordination numbers in the range of 6-8 (for the free ion). This is already accounted for in the association free energy calculations performed using molecular dynamics, where the Ca-water coordinate number is explicitly included as a collective variable. For the ab initio molecular quantum mechanical data, the spontaneous exploration of the water coordination number was indeed initially restricted to 6-fold coordination. To address this point, all calculations have been repeated for 7-fold coordination of calcium by water in order to determine the sensitivity to this choice (8-fold coordination is unlikely to make a substantial contribution, as this has been shown to lie higher in free energy).³ As now described in the manuscript and SI, the inclusion of both 6- and 7-fold coordination leads to an increase in the free energy difference for ion pairing, though the value remains substantially smaller than the current literature value. Indeed, the change to 7-fold coordination for calcium brings the computed value into much better agreement with our proposed new equilibrium constants, and so we thank the reviewer for this helpful suggestion.

3. Some related literatures should be cited for comparison, e.g., Tan et al., Cryst. Growth Des. 2020, 20, 4561.

We agree that it is appropriate to compare values with the prior literature. Unfortunately, most previous simulation work has only tended to consider HPO_4^{2-} and not PO_4^{3-} limiting the extent of full comparison to the works already cited. However, we thank the referee for pointing out that this work, whose primary focus is on collagen, but does contain an estimate of the free energy of ion pairing for CaPO_4^- and therefore should indeed be cited. Unfortunately, the value contained in this reference cannot be reliably compared against our simulations or experiment, because the free energy is not corrected to standard conditions by alignment with

the long-range asymptote and normalisation to standard volume. In addition, it was only computed using distance as a collective variable, rather than cation-water coordination number as well, which may explain some of the unexpected features in their free energy profile that doesn't conform to the expected shape of minima for the contact, solvent-shared and solvent-separated ion pair, with a smooth longer-range tail. Furthermore, the force field used was a rigid ion model (INTERFACE FF) which is less sophisticated than the polarisable model used here and has less extensive calibration for this system. Discussion of the ion pairing free energy of Tan et al., which is more exothermic than the current value, and why it should not be considered more reliable has now been added to the manuscript in both the Results and Methods sections (the latter discussion relating to the more technical aspects of the methodology).

4. Forms for some references should be carefully checked and corrected. Moreover, Ref21 lacks page number.

Reference formatting has been updated. For Ref. 21 the page number has been added.

5. Line 252, "... have a developed an improved..." should be "... have developed an improved..."

We apologise for the typographical error. This has been corrected as suggested.

Other changes:

As per the author guidelines, we have included the Methods section in the main text. Also, figure references and other formal aspects were adapted to meet these requirements.

References

1. McDonogh DP, Kirupanathan P, Gebauer D. Counterintuitive Crystallization: Rate Effects in Calcium Phosphate Nucleation at Near-Physiological pH. *Cryst. Growth Des.* **23**, 7037-7043 (2023).
2. Avaro JT, Wolf SL, Hauser K, Gebauer D. Stable Pre-nucleation Calcium Carbonate Clusters Define Liquid-Liquid Phase Separation. *Angew. Chem.* **59**, 6155-6159 (2020).
3. Baer MD, Mundy CJ. Local aqueous solvation structure around Ca²⁺ during Ca²⁺... Cl⁻ pair formation. *J. Phys. Chem. B* **120**, 1885-1893 (2016).

Reviewers' Comments:

Reviewer #1:

Remarks to the Author:

Thanks so much to the great efforts tried by the author to address my concerns. Most of my concerns have been solved properly. I have one more suggestion for Fig. S12. The value of calcium ion activities are so different at different pH, if author can also add a similar plot of percentage of calcium ions activities over the added calcium ions, that would be much easier to compare with Fig. S13.

Reviewer #2:

Remarks to the Author:

Overall, I like the current revised version. The authors have addressed all my concerns. It should be accepted for publication in current form.

Response to Reviewer 1:

Reviewer: "Thanks so much to the great efforts tried by the author to address my concerns. Most of my concerns have been solved properly. I have one more suggestion for Fig. S12. The value of calcium ion activities are so different at different pH, if author can also add a similar plot of percentage of calcium ions activities over the added calcium ions, that would be much easier to compare with Fig. S13."

Response: This is another great suggestion by the Reviewer. A plot of the ratio of the activity of free Ca^{2+} ions to the activity of added Ca^{2+} ions as calculated using the literature constants along with the predictive calculation using the literature constants has been added to Supplementary Figure 12. Additionally, the fraction of bound Ca^{2+} for the same as above has been added to Supplementary Figure 13. This will allow readers to compare the values within one chart. This also demonstrates the good agreement between PHREEQC model, experimental measurements and the predictive calculation.

Response to Reviewer 2:

Reviewer: "Overall, I like the current revised version. The authors have addressed all my concerns. It should be accepted for publication in current form.

Remarks on code availability:

The README file contains enough information. Basically, this code does not need to be installed for further running. The current codes were written using python script. All needed data were included."

Response: We appreciate the positive feedback and thank the Reviewer for their reports.